# The Role of Neutrophils in Lower Limb Peripheral Artery Disease: State of the Art and Future Perspectives

**DOI:** 10.3390/ijms24021169

**Published:** 2023-01-06

**Authors:** Giacomo Buso, Elisabetta Faggin, Nathalie Rosenblatt-Velin, Maxime Pellegrin, Silvia Galliazzo, Luca Calanca, Marcello Rattazzi, Lucia Mazzolai

**Affiliations:** 1Angiology Division, Heart and Vessel Department, Lausanne University Hospital, University of Lausanne, 1011 Lausanne, Switzerland; 2Department of Medicine, University Hospital of Brescia, 25123 Brescia, Italy; 3Department of Medicine—DIMED, University of Padua, 35128 Padua, Italy; 4Department of Internal Medicine Unit, Ospedale S. Valentino, 31044 Montebelluna, Italy; 5Medicina Interna I, Ca’ Foncello University Hospital, 31100 Treviso, Italy

**Keywords:** peripheral artery disease, atherosclerosis, neutrophil activation, neutrophil extracellular traps, inflammasome

## Abstract

In recent years, increasing attention has been paid to the role of neutrophils in cardiovascular (CV) disease (CVD) with evidence supporting their role in the initiation, progression, and rupture of atherosclerotic plaque. Although these cells have long been considered as terminally differentiated cells with a relatively limited spectrum of action, recent research has revealed intriguing novel cellular functions, including neutrophil extracellular trap (NET) generation and inflammasome activation, which have been linked to several human diseases, including CVD. While most research to date has focused on the role of neutrophils in coronary artery and cerebrovascular diseases, much less information is available on lower limb peripheral artery disease (PAD). PAD is a widespread condition associated with great morbidity and mortality, though physician and patient awareness of the disease remains low. To date, several studies have produced some evidence on the role of certain biomarkers of neutrophil activation in this clinical setting. However, the etiopathogenetic role of neutrophils, and in particular of some of the newly discovered mechanisms, has yet to be fully elucidated. In the future, complementary assessment of neutrophil activity should improve CV risk stratification and provide personalized treatments to patients with PAD. This review aims to summarize the basic principles and recent advances in the understanding of neutrophil biology, current knowledge about the role of neutrophils in atherosclerosis, as well as available evidence on their role of PAD.

## 1. Introduction

Lower limb peripheral artery disease (PAD) is a widespread condition affecting more than 200 million people worldwide, including 40 million people living in European countries [1]. It consists in a progressive narrowing of the arteries of the legs, which can cause reduced blood flow and oxygen delivery to the lower limbs impairing muscle function and overall quality of life. The hallmark symptom of PAD is intermittent claudication, which occurs in approximately 30% of affected individuals [2]. Patients with PAD may also present with severe local complications, or major adverse limb events (MALE), including lower limb amputation and acute limb ischemia (ALI). Furthermore, PAD is a marker of systemic atherosclerosis in other vascular territories and is therefore associated with an increased risk of major adverse cardiovascular (CV) events (MACE), such as myocardial infarction, stroke, and CV death [3].

As for coronary artery disease (CAD) and cerebrovascular disease (CeVD), PAD is most often secondary to atherosclerosis [4]. From a pathophysiological perspective, chronic inflammation has been widely recognized as a hallmark of CV disease (CVD), with evidence supporting a role of inflammation in the initiation, progression, and rupture of atherosclerotic plaques [5]. In recent years, increasing attention has been paid to the role of neutrophils in this setting.

Neutrophils are the most abundant effector cells of the innate immune system, accounting for 75% of the total population of circulating leucocytes in humans [6]. The powerful weapons of neutrophils include reactive oxygen species (ROS), neutrophilic-derived microparticles (NMPs), lipid mediators such as leukotriene B4, as well as factors stored in neutrophilic granules. This toolkit includes metalloproteinase (MMP)−8 and −9, myeloperoxidase (MPO), neutrophil gelatinase-associated lipocalin (NGAL), neutrophil elastase (NE), proteinase 3 (PR3) cathepsin G, and alarmins (such as defensins and cathelicidins) [7].

Neutrophils have long been considered a homogeneous, terminally differentiated cell population with short half-life, vast turnover, and poorly organized activity. Nonetheless, recent research has challenged this paradigm, revealing an extreme diversity of human neutrophils in vivo [8] and novel organized cellular functions such as neutrophil extracellular traps (NETs) generation and inflammasome activation, which have been shown to be involved in a variety of clinical conditions, including CVD [9].

This review aims to summarize the most recent advances in the understanding of neutrophil biology, current knowledge on the role of neutrophils in atherosclerosis, as well as available evidence in the specific setting of PAD. Lastly, we propose some potential lines of research with therapeutic implications in this field.

## 2. Data Sources and Research Strategy

We searched in PubMed, EMBASE and Google Scholar English-language articles published up to 30 September 2022 including the following MeSH terms: “neutrophil (and related terms: neutrophil biology, neutrophil activation, neutrophil activity, neutrophil inflammasome) and atherosclerosis”, “neutrophil and PAD (and related terms: peripheral artery disease, lower limb extremity artery disease, LEAD, acute limb ischemia)”, “neutrophil-to-lymphocyte ratio and PAD”, “neutrophil gelatinase-associated lipocalin (and related terms: lipocalin, NGAL) and PAD”, “myeloperoxidase (and related term: MPO) and PAD”, “matrix metalloproteinase (and related term: MMP) and PAD”, “neutrophil extracellular trap and PAD”, “neutrophil inflammasome and PAD”. Case reports, case series, commentaries, letters to editors, original and review articles were considered. Two coauthors independently evaluated title and abstract for inclusion, double-checking for duplication and extracting the data using the text, tables, and figures of the original published articles.

## 3. Issues Challenging the Old Conception of Neutrophil Biology

### 3.1. Neutrophil Extracellular Traps Generation

Upon proper stimulation, neutrophils can generate NETs, a process known as NETosis [10]. NETs were first discovered by Brinkmann et al. in 2004 and were described as an extracellular fibrillar network consisting of chromatin and granular proteins through which neutrophils are able to immobilize and capture bacteria, fungi, and other pathogens [11].

Two main different processes of NETs generation have been identified so far. The best known is late suicidal NETosis. Different from necrosis or apoptosis, it is a well-orchestrated process of cell death during which neutrophils undergo substantial changes in their morphology through precise steps. This process requires strong and lasting stimuli, resulting in exaggerated ROS production via reduced nicotinamide adenine dinucleotide phosphate (NADPH) oxidase. The importance of NADPH oxidase in this process is suggested by the absence of NETosis in patients with inherited disorders involving NADPH oxidase defects and after using inhibitors of NADPH oxidase and H2O2 degradation [12], although NADPH-independent pathways of NETosis have been described [13]. ROS activate protein-arginine deaminase 4 (PAD4) [11], which leads to hypercitrullination of the arginine residues of histones H3 and H4 resulting in loss of positive charge and subsequent chromatin decondensation [14]. PAD4 seems to be crucial for the initiation of NETosis, as neutrophils from PAD4 deficient mice have been reported to undergo a lower degree of NETosis [15]. Nonetheless, PAD4-independent mechanisms have also been described [16], and further studies have shown diminished, but not abolished, NETosis detected with an anti-histone H3 antibody in PAD4-deficient mice [17]. Besides activating PAD4, ROS lead to MPO and NE release from azurophilic granules within cells and their migration to the nucleus, where they further cleave histones and synergistically facilitate chromatin unfolding [18]. Finally, chromatin filaments enriched by enzymes and histones are released into the extracellular space after nuclear membrane disruption, 45–240 min after neutrophil activation [18]. The presence of granular proteins may be crucial for NETosis, as NE knockout mice, patients with MPO deficiency, as well as MPO inhibition in experimental studies are associated with reduced amounts of NETs [18,19].

A rapid, vital form of pathogen-induced NETosis has also been reported [20]. In this process, which allows neutrophils to maintain their phagocytic capacity, the plasma membrane is preserved and chromatin is released via nuclear-envelope blebbing and vesicular export a few minutes after PAD4 activation [21,22,23].

Besides bacteria or fungi, several stimuli, including lipopolysaccharide (LPS), interleukin (IL)-8, activated platelets, cholesterol and monosodium urate crystals, as well as several autoantibodies can induce NETosis and amplify sterile inflammation [24,25,26]. Accordingly, increasing evidence has emphasized the role of NETs in a large spectrum of noninfectious diseases, including autoimmune disorders [27], cancer [28], venous thromboembolism [29,30], as well as atherosclerotic diseases [31,32,33].

### 3.2. Inflammasome Activation

Inflammasomes are intracellular, multiprotein complexes expressed primarily in myeloid cells, such as monocytes and macrophages [34], but also in dendritic cells and neutrophils [35].

Among the various inflammasomes described, the nucleotide-binding oligomerization domain, leucine rich repeat, and pyrin domain-containing protein (NLRP)3 inflammasome is the most studied, although many other varieties have been described so far, including the NLRP12 and the absent-in-melanoma (AIM)2 inflammasome [36].

These complexes trigger local or systemic inflammatory responses upon danger stimuli through the formation of a caspase-1-activating platform and the subsequent production of pro-inflammatory cytokines belonging to the IL-1 family, such as IL-1β and IL-18 [37]. Caspase-1 also cleaves a pore-forming protein, gasdermin D, to release its N-terminal fragment, which permeabilizes the cell membrane causing pyroptosis [38]. Additionally, through other caspases (caspase-11 in mice and caspase-4 and 5 in humans), a noncanonical inflammasome monitors cytosolic bacteria-derived LPS [39,40]. Furthermore, inflammasomes constitute a potential crucial regulator of myeloid lineage differentiation through modulation of cholesterol and glucose metabolism within hematopoietic stem and progenitor cells, a critical process during trained immunity [41]. In particular, inflammasome-derived caspase-1 leads to transcription factor GATA cleavage upon chronic inflammation, resulting in increased neutrophil production at the expense of erythroid differentiation [42], which may partly explain the prevalent combination of neutrophilia and anemia in patients with chronic inflammation.

Macrophages and dendritic cells have long been considered responsible for inflammasome activation and IL-1β production, whereas neutrophils were thought to be solely targets of pro-inflammatory cytokines. However, in recent years it has been shown that neutrophils are capable of activating several inflammasomes and have been suggested as the main source of mature IL-1β during bacterial infection, at least in its early stages [36].

Similar to macrophages, neutrophils express inflammasome components in the cytoplasm [35,43]. However, these components also seem to localize near the secretory vesicles of neutrophils [35,44], suggesting a possible role in the regulation of secretory vesicles and granules. Furthermore, translocation of these components to the surface of neutrophils may be involved in pathogen recognition and uptake [45,46], while the release of inflammasome components in the extracellular space has been hypothesized to provide endogenous danger signals to further amplify the inflammatory response [47].

Importantly, a crosstalk between the inflammasome and NETs was described. In both murine models and humans, neutrophils have been found to elicit noncanonical inflammasome activation and gasdermin D cleavage by caspase-11 to activate NETosis in response to cytosolic bacteria [48], whereas chemical inhibition of gasdermin D avoids neutrophil inflammasome activation and blocks both IL-1β release and NETosis [49]. In a recent study using a mouse model, PAD4 was found to be necessary for optimal NLRP3 inflammasome assembly in neutrophils. Furthermore, genetic ablation of NLRP3 signaling and pharmacological inhibition of NLRP3 in either mouse or human neutrophils resulted in impaired NETosis [50]. On the other hand, NETs can also trigger NLRP3 inflammasome activation and IL-1β release, whereas NET digestion by DNase I has been shown to attenuate NLRP3 inflammasome activation [51].

This interplay between inflammasome and NETs has been described in several clinical conditions, including systemic lupus erythematosus [52], severe asthma [53], and venous thromboembolism [50,54].

### 3.3. Neutrophil Heterogeneity and Plasticity

Unlike other cells in the hematopoietic system, such as lymphocytes and macrophages, neutrophils have long been considered to lack heterogeneity, given their short half-life and their reduced transcriptional activity. Instead, more recent research has demonstrated some capacity for de novo protein synthesis by mature neutrophils, involving both cytokines and membrane receptors. This has allowed neutrophils to be classified into different subpopulations under both physiological and pathological conditions [55], including proangiogenic neutrophils (PAN), neutrophils in reverse migration, differentiation cluster (CD)177+ cells, and neutrophils with olfactomedin 4 (OLFM4)+ [56].

The PAN population, which represents only approximately 3% of total circulating neutrophils in humans, is retained at sites of hypoxia, where it releases high amounts of MMP-9 contributing to extracellular matrix remodeling and restoration of blood flow after ischemia [57]. Although neutrophil recruitment has been traditionally considered unidirectional, animal models suggest that these cells can also migrate from the tissues to the blood and even to other tissues, which can facilitate their clearance or, conversely, provoke secondary tissue damage. This feature, which was first described in the context of ischemia/reperfusion injury [58], is associated with a peculiar pro-inflammatory phenotype characterized by increased expression of intracellular adhesion molecule (ICAM)-1, decreased expression of C-X-C motif chemokine ligand (CXC) receptor (CXCR)1, and increased ROS [59]. CD177+ cells represent approximately half of the total circulating neutrophils under physiological conditions and have been suggested to be involved in anti-neutrophil cytoplasmic antibody (ANCA)-dependent vasculitis [60], as might also be the case with OLFM4+ cells. Interestingly, compared with OLFM4+ neutrophils, OLFM4- cells produce distinct NETs in vitro [61] and may be effective than positive neutrophils in infectious diseases [62,63].

### 3.4. Regulation of Neutrophil Metabolism

Cellular lipid and glucose metabolism are crucial regulators of granulopoiesis and neutrophil function in physiological conditions (e.g., during infections). However, metabolic dysregulation sustains chronic inflammation and accelerate atherosclerosis in CVD. Defective cholesterol efflux increases accumulation of cholesterol in the cell membrane inducing hematopoietic stem and progenitor cells proliferation and mobilization and a differentiation toward the myeloid cell lineage in mice [64]. On the other hand, hyperglycemia has been shown to increase granulopoiesis and impair atherosclerosis resolution in mice [65]. It is also capable of promoting both myelopoiesis [65] and IL-6-mediated thrombocytosis [66] via the release from neutrophils of the alarmins S100A8 and S100A9 and their interaction with myeloid progenitor cells and Kupffer cells. This phenomenon contributes to atherogenesis and atherothrombosis [9].

Neutrophil plasticity and cellular metabolism are promising topics to be further explored in the future in the setting of atherosclerotic diseases.

## 4. Neutrophils, Endothelial Dysfunction, and Atherosclerosis

### 4.1. Neutrophils Promote the Initiation of Atherosclerosis

Atherosclerosis is by far the most important disorder underlying CVD and is therefore responsible for substantial disability and death worldwide [4]. The role of neutrophils in atherogenesis is summarized in Figure 1. 

During its initial stages, unbalanced blood lipid profiles, abnormal shear forces, and local release of pro-inflammatory cytokines cause endothelial layer dysfunction and subsequent disruption of the underlying extracellular matrix. This enables immune cells adhesion and transmigration and accumulation of low-density lipoprotein (LDL) and oxidized LDL (ox-LDL) within the basal aspect of the artery, leading to fatty streak formation [67]. Platelet-derived chemokines, such as C-C motif chemokine ligand (CCL)5, promote neutrophil activation and recruitment. Moreover, hypercholesterolemia induces neutrophilia and increases neutrophil mobilization through CXCL1 production [68,69,70], while decreasing neutrophils clearance in the bone marrow and enhancing their aging through reduced CXCL12 levels [71]. The resulting low-grade inflammation further activates the endothelium and increases ox-LDL deposition [70]. In turn, activated neutrophils secrete chemotactic proteins like cathepsin G and cathelicidin that promote monocyte recruitment [72]. These two mediators, as well as other molecular complexes including α-defensin and platelet-derived CCL5, localize on endothelial cells (ECs) surface and induce firm monocyte adhesion in animal models of vascular inflammation [73,74,75]. Other granule proteins, such as azurocidin and PR3, increase endothelial expression of adhesion molecules and regulate EC permeability [76]. Neutrophils and their granular proteins can also modify macrophages towards a pro-inflammatory profile [77]. NMPs released into the extracellular space can further activate immune cells and promote cytokine release and adhesion protein expression in ECs [78,79,80,81,82]. For example, a disintegrin and MMP with thrombospondin motifs (ADAMTS) can promote tumor necrosis factor (TNF)-α processing and increase inflammation [83,84].

Beyond degranulation and NMPs release, neutrophils may also contribute to atherogenesis through NETs generation. The role of NETs in venous thrombosis is well known, whereas its contribution to atherosclerosis is still under investigation. NETs have been described within atherosclerotic lesions in animal models and patients [31,32]. Moreover, PAD4 inhibition in apolipoprotein E−/− mice resulted in protection from atherosclerosis [85], whereas mice deficient in NE and PR3 displayed reduced growth of atherosclerotic lesions [86]. As reported above, a large number of stimuli have been shown to induce NETosis. In the setting of atherosclerosis, platelets can activate neutrophils and promote NETs generation through CCL5 and CXCL4 [31,87]. Once released in the extracellular space, NETs containing DNA-cathelicidin-related antimicrobial peptide (CRAMP) complexes stimulate plasmacytoid dendritic cells to produce interferon (IFN)-α [88]. Additionally, NETs can induce macrophages to produce IL-1β and IL-18 via the NLRP3 inflammasome [86], promoting atherosclerotic lesions in LDL receptor (LDLr)-deficient female mice [64]. Intriguingly, recent research has found that NETs induced by nicotine, a common pro-atherosclerotic agent, negatively regulate autophagy in macrophages through Beclin-1 suppression, which enhances inflammasome activity [89].

Of note, previous studies have shown increased expression of NLRP3 inflammasome, IL-1β, and IL-18 in patients with CAD [90,91,92], suggesting a role of such mechanisms in atherogenesis in humans as well, although further research is needed in this regard.

### 4.2. Neutrophils Promote Atherosclerotic Plaque Progression and Instability

The role of neutrophils in plaque progression and rupture is summarized in Figure 1. Within the media layer of arteries, vascular smooth muscle cells (VSMCs) produce indoleamine dioxygenase in response to lymphocyte Th1-derived IFN-γ in an attempt to restrain the spread of intimal inflammation across the vessel wall. Thereafter, VSMCs begin to proliferate and migrate from the media to the intima, where they participate in the formation of the fibrous cap that covers the inflammatory core of the plaque [93].

Collagen degradation and VSMCs death result in fibrous cap thinning and subsequent plaque rupture. Studies on human samples of thin fibrous cap atheroma have shown higher neutrophil counts in rupture-prone lesions than in stable ones [94]. VSMC death is induced by neutrophil-derived MMPs via degradation of the endothelial layer. Moreover, activated VSMCs stimulate NETs generation through the secretion of CCL7, and histone H4 has been shown to have cytotoxic effects in VSMCs [17]. Extracellular double-stranded DNA, another well-known component of NETs, can be sensed by macrophages through the AIM2 inflammasome [95] and colocalizes in mature atherosclerotic lesions with the noncanonical inflammasome AIM2 [96], whereas mice deficient in both AIM2 and apolipoprotein E display reduced IL-1β production and plaques with a less unstable phenotype [95].

Besides plaque rupture, neutrophils can trigger luminal ECs desquamation, a process known as plaque erosion, which has gained great attention in recent years. In human samples, neutrophils colocalize with Toll-like receptor 2 (TLR2) patches on ECs [97]. Neutrophils can stimulate TLR2, leading to stress and apoptosis of ECs, as evidenced in both human studies and animal models [97,98]. Furthermore, research in atherosclerotic mice has provided genetic evidence for the involvement of NETs during endothelial erosion, as PAD4 deficiency or treatment with DNase I reduced EC apoptosis and discontinuity in these mice [99].

### 4.3. Neutrophils Promote Atherothrombosis

While the role of neutrophils in venous thromboembolism has been largely investigated [29,30,100], less evidence exists in the setting of atherothrombosis. Neutrophils have been detected in coronary thrombi [33,101,102], as well as in surgical thrombectomies and abdominal aortic aneurysms [103]. Analysis of thrombi from patients with ST-segment elevation myocardial infarction undergoing primary percutaneous coronary intervention documented NETs presence. The authors also found a positive correlation between infarct size and NETs burden, whereas ex vivo addition of DNase accelerated thrombus lysis [33]. Neutrophils and NETs were also abundant in cerebral thrombi in the setting of ischemic stroke [104]. Mechanistically, research has revealed that activated platelets can induce NETs generation through high-mobility group box 1 (HMBG1) presentation [105], which played a role in blood clot formation in a mouse model of venous thrombosis [106]. Cathelicidins have also been found in human and murine arterial thrombi, and CRAMP depletion reduced the development of arterial thrombosis in a mouse model [107]. In a recent study, liver-to-lung microembolic NETs were found to be capable of promoting gasdermin D-dependent inflammatory lung injury in the setting of sickle cell disease [108].

The possible role of inflammasomes in thrombosis has been outlined in a recent study showing that NLRP3 inflammasome activation and subsequent IL-1β production enhance venous thrombosis in response to hypoxia [109]. Furthermore, NLRP3 protein levels in peripheral blood monocytes were found to be increased in patients with acute coronary syndrome and directly correlated with disease severity [110], whereas the CANTOS (Canakinumab Anti-Inflammatory Thrombosis Outcomes Study) trial clearly demonstrated the benefits of canakinumab, a monoclonal antibody targeting IL-1β, in terms of MACE reduction in patients with a previous myocardial infarction [111]. However, further research may better delineate the relevance of neutrophil-derived inflammasomes in the setting of atherothrombosis.

## 5. The Established Role of Neutrophils in Peripheral Artery Disease

So far, substantial evidence has been provided for the role of neutrophils in CAD, CeVD, and venous thromboembolism [9,13,56,112,113,114,115], whereas relatively little evidence exists regarding PAD. In particular, no significant insight has been provided so far on the role of neutrophils in the etiopathogenesis of PAD and its complications. Accordingly, no distinctive features have been clearly highlighted compared with the other two common CVDs. Instead, most of the published research on PAD has focused on the diagnostic and prognostic ability of biomarkers of neutrophil activity that had already been extensively explored in other CVDs. The results of these studies are reported below and summarized in Table 1.

### 5.1. Neutrophil-To-Lymphocyte Ratio

Neutrophil-to-lymphocyte ratio (NLR) is one of the most studied biomarkers of systemic inflammation in a wide range of pathological conditions, including CVDs [163].

There are several possible explanations for the relationship between elevated NLR and atherosclerosis. Unlike neutrophils, regulatory CD4+ T-helper cells, a subclass of lymphocytes, may have an inhibitory effect on atherosclerosis [164]. Rather than expressing the abundance of circulating neutrophils, the NLR could thus reflect a profound imbalance in the immunological response, with effectors cells predominating over regulatory cells [165,166]. Moreover, previous studies have shown that NLR predicts levels of C-reactive protein (CRP) and high sensitivity CRP (hs-CRP) [167], which are strongly associated with the risk and prognosis of CVDs [168].

As for PAD, NLR levels were found to be significantly higher in subjects with an ankle-brachial index (ABI) ≤ 0.9 than in those with an ABI 0.9–1.4 [117], as well as in patients with PAD on angiography compared with those without PAD [116], suggesting that this index may be used as a simple and reproducible marker of disease. The NLR significantly correlated with early stages of PAD assessed by arterial ultrasound [169]. In patients referred for lower limb angiography, increased NLR values were independently associated with the presence of multi-level PAD [119] and independently predicted the presence of severe and complex PAD as assessed by the Trans-Atlantic Inter-Society Consensus-II (TASC-II) classification [118]. In two retrospective cohorts of patients with PAD, high NLR levels were independently associated with the presence of critical limb-threatening ischemia (CLTI) [121,122]. Conversely, a retrospective analysis of patients undergoing distal autologous bypass procedures found no difference in terms of NLR between patients with CLTI and disabling claudication, although ischemic ulcers were more frequent, more severe, and pedal infections deeper in patients in the highest NLR quartile [123].

Besides being a disease with local consequences, PAD is also a marker of systemic atherosclerosis. In particular, in patients with PAD, CAD is common and is a major cause of death [170]. In a cross-sectional study, patients with PAD were divided in two groups based on the presence or absence of concomitant CAD, defined as stenosis of at least 30% on conventional computed tomography angiography. Patients with both diseases had higher levels of NLR than those with PAD alone, whereas NLR levels were significantly associated with multi-bed vascular disease on multivariate analysis [120]. In a retrospective study on patients with symptomatic PAD, the majority of whom presented had stable claudication, a NLR > 3 independently predicted long-term CV death [171]. In a large cohort of patients undergoing revascularization procedures (of whom approximately 60% underwent endovascular treatment), pre- and post-operative NLR levels were strongly associated with several adverse outcomes, including in-hospital death and prolonged length of hospitalization [124]. In patients undergoing elective surgery in two tertiary vascular units, a pre-operative NLR > 5 was an independent predictor of 2-year mortality [126]. Similar findings have been described for longer follow-ups [127]. NLR levels also predicted target artery restenosis after infra-inguinal angioplasty with [128] or without [129] stent implantation. In patients with CLTI, pre-operative NLR levels were independently associated with mortality at 6 and 12 months and major amputations at 6 months [130,172]. In two other studies on patients with CLTI undergoing vascular surgery, multivariate analysis confirmed that pre-operative NLR > 5 was independently associated with 12-month [131] and 5-year amputation-free survival [132]. Furthermore, in a retrospective study on patients undergoing percutaneous interventions of femoro-popliteal arteries, most of whom had CLTI, increasing NLR levels correlated with increasing 30-day mortality rates and amputation-free survival at 4-year follow-up [125]. More recently, pre-operative NLR levels were found to be strongly predictive of patency loss, amputation, and mortality 12 months after all types of revascularization for femoro-popliteal artery disease [133]. In patients undergoing infra-inguinal bypass grafting, NLR levels predicted graft failure, defined as graft occlusion or ipsilateral amputation [134]. Furthermore, in a retrospective analysis of patients treated with distal autologous bypass, high pre-operative NLR values were independent predictors of 2-year patency loss, MALEs, and amputations [123]. NLR levels ≥ 5.25 were also independently associated with 12-month death in patients with CLTI, including those undergoing infra-popliteal percutaneous intervention [135,173]. In a retrospective study on patients with CLTI unsuitable for revascularization and receiving standard pharmacological treatment including low molecular weight heparin, aspirin, statins, iloprost infusion, and a standard pain medication protocol, lower pre-treatment levels of NLR were associated with subsequent improvement in ischemic pain and healed ulcers [136]. Other authors showed similar findings [138]. In another prospective study on patients with CLTI who received conservative therapy, post-treatment NLR levels were independent predictors of amputation on multivariate analysis [137].

As for ALI, pre-operative NLR values were independently associated with 30-day death and amputation in several retrospective studies on patients undergoing lower limb revascularization for ALI [139,140,142,143]. Similar results were found for longer follow-ups [141,142,174].

These studies have some important limitations. The first is the use of different cutoff values in the various studies. Moreover, very little evidence has been provided on normal NLR values in the general population [175], whereas no study validated the normality value for specific populations. The use of this biomarker has not yet been implemented in clinical practice, and further, solid evidence in this regard is needed.

### 5.2. Myeloperoxidase

Among the extracellular proteases stored in neutrophilic granules and released by neutrophils upon activation, MPO has been linked particularly to atherogenesis. MPO is a heme-containing peroxidase stored in azurophilic granules and released into the extracellular space during degranulation [176]. MPO is found in atherosclerotic plaques [177], and further research has demonstrated that MPO-containing macrophages are abundant in vulnerable and ruptured plaques rather than in earlier stages of atherosclerosis [178]. Moreover, individuals with MPO deficiency have been found to be at lower risk of CVD [179], whereas genetic polymorphisms of MPO deficiency have been shown to be associated with an increased risk of CAD [180]. Higher serum levels of MPO have been associated with both the presence [181] and severity [182,183] of CAD on angiography and seem to be predictive of adverse outcomes in both patients presenting to the emergency department with chest pain [184] and those with an established diagnosis of CAD [183].

In a study on patients with PAD treated with endovascular therapy using a filter device to prevent distal embolization, histologic, immunohistochemical, and immunofluorescence analyses revealed the presence of inflammatory cells, mainly CD68-positive cells that were also positive for MPO, in nearly half of the debris particles [185]. Higher blood MPO levels have been reported in patients with PAD than in healthy subjects [144,186], although others have found opposite results [186]. Notably, an inverse association between MPO levels and ABI has been described in non-Hispanic white individuals, even after additional adjustment for potential confounders, such as diabetes, smoking status, blood cholesterol, waist circumference, hypertension, history of myocardial infarction or stroke, statins and aspirin use, as well as CRP [145,147]. In a large cohort of patients with chronic kidney disease, MPO was independently associated with PAD risk during a 6.3-year follow-up [146].

Higher MPO levels are also associated with adverse events in patients with PAD. In a prospective study on patients with symptomatic PAD, MPO levels ≥ 183.7 pM were predictive of myocardial infarction or stroke during a median follow-up of 17.5 months. The results remained consistent when MPO was used as a continuous variable. Intriguingly, CRP was not predictive of MACE in this study [149], suggesting that MPO might have greater predictive power in terms of MACE in patients with PAD. Another study found that patients with MPO levels > 115 ng/mL had a significantly higher risk of MACE than those with lower levels, and the risk was particularly high in both males and smokers [150]. In a recent prospective cohort study on patients with PAD, CAD, or both, rates of all-cause mortality and MACE increased with increasing MPO levels, whereas MPO was associated with rates of limb ischemia and further revascularization in patients with PAD. Furthermore, higher MPO levels have been reported in patients with multi-bed vascular disease [148]. Recent research has shown that plasma MPO indicates high-density lipoprotein dysfunction and thus suggested using this marker in risk stratification in patients with PAD [187]. In another study on CAD patients with or without PAD, higher levels of MPO were reported in the femoral circulation than in the coronary circulation in subjects with multi-bed vascular disease. These findings did not apply to CRP. Intriguingly, coronary artery endothelial dysfunction correlated with the transfemoral gradient of MPO, whereas serum from affected limbs of patients with multi-bed vascular disease induced, in vitro, greater release of monocyte chemotactic protein 1 from human coronary artery ECs than serum from aorta. These findings may suggest a causal role of PAD in cardiac complications [188].

Interestingly, in leptin receptor-deficient (db/db) mice subjected to hindlimb ischemia through ligation of the femoral artery, MPO inhibition improved hindlimb blood flow at seven and 14 days, reduced generation of toxic oxidants and neutrophil infiltration into the hindlimbs, and improved the ability of the hindlimb extracellular matrix to support EC proliferation and migration [189].

Again, a major limitation in the use of circulating MPO as a biomarker is the absence of research validating normal values in general and in specific populations. Further evidence is needed in this regard.

### 5.3. Neutrophil Gelatinase-Associated Lipocalin

NGAL, also known as human lipocalin or lipocalin 2, is a small protein stored in specific granules in neutrophils. NGAL has been extensively studies as a biomarker of acute kidney injury [190,191], although recent studies have also highlighted its role in CVDs [192]. NGAL has been hypothesized to be a marker of atherosclerosis, as it is expressed in areas of high proteolytic activity in atherosclerotic plaques [193]. High plasma levels of NGAL correlate with CAD severity [194] and independently predict all-cause mortality and MACE in patients with ST-segment elevation myocardial infarction after primary percutaneous coronary intervention [195]. In a multicenter prospective study, higher blood levels of NGAL were reported in patients with arterial aneurysms compared with healthy controls, particularly in those with ruptured ones. In these patients, Western blot analysis revealed a particularly high NGAL expression in aortic tissues collected during surgery [196]. NGAL has also been associated with intra-plaque hemorrhages in carotid endarterectomy samples [197]. In a study on patients with diabetes and poor glycemic control without known CVD, increased NGAL at baseline was independently associated with CV events at follow-up [198]. Similar findings have been reported in the setting of chronic kidney disease [199].

In the context of atherosclerotic diseases, the role of NGAL seems to involve its interaction with MMP-9 [200,201], which is stored in the gelatinase granules of neutrophils. MMPs are a multi-gene family of zinc- and calcium-dependent endopeptidases involved in remodeling of the extracellular matrix through the proteolysis of its components. The NGAL/MMP-9 complex inhibits MMP-9 degradation and prolongs its proteolytic activity [202]. This, in turn, increases the degradation of extracellular matrix and basement membranes, leading to atherosclerotic plaques instability. In patients undergoing surgery for infra-renal abdominal aortic aneurysms, NGAL/MMP-9 complexes were detected both in the thrombus and at the interface between the thrombus and the underlying wall [155]. Both circulating levels of MMP-9 and NGAL are increased in patients with ALI compared with healthy volunteers, and particularly in those undergoing fasciotomy for compartment syndrome [153].

NGAL has been detected in human femoral plaques in regions rich in inflammatory cells and high serum levels of NGAL increase the risk of CV death and MALE in both patients with symptomatic PAD and those with early stage disease [203]. NGAL gene expression was also significantly higher in circulating extracellular vesicles from patients with symptomatic PAD than in healthy subjects, and these extracellular vesicles have been hypothesized to be associated with a prothrombotic state [154]. NGAL levels were found to be significantly higher in patients with PAD, and particularly in those with CLTI [152]. In a recent, prospective study, urinary NGAL (uNGAL) was measured in patients with PAD and healthy controls and normalized for urinary creatinine (uCr). PAD patients had 2.30-fold higher levels of uNGAL/uCr, whereas multivariate Cox analysis showed that uNGAL/uCr levels were independent predictors of PAD status worsening, defined as a drop in ABI > 0.15, and MALEs, defined as the need for surgical revascularization or amputations [151].

Although causality is not explicitly demonstrated, the above findings might suggest a pro-atherogenic action of NGAL through NGAL/MMP-9 complexes. However, further studies are needed to confirm the hypothesis and explore the role of NGAL specifically in PAD.

### 5.4. Matrix Metalloproteinases

Previous evidence has shown that circulating levels of MMP-9 are increased in patients with PAD, and particularly in those with ALI [153] and CLTI [156]. Furthermore, MMP-9 seems to be upregulated in skeletal muscle capillaries of patients with ALI and CLTI compared with healthy subjects [204]. In patients with CLTI undergoing infra-popliteal vein graft and midfoot amputation, post-operative increase in MMP-9 levels was found to be associated with graft occlusion at follow-up [159]. In a study using a rat model, the nonselective MMP-inhibitor doxycycline reduced intimal hyperplasia and MMP-9 activity in the arterial wall [205]. However, after femoral artery ligation, bone marrow cell-specific loss of MMP-9 in mice led to increased necrotic and fibro-adipose tissue, which showed the strongest correlation with poor perfusion recovery [206]. Of note, MMP-9 may actually promote angiogenesis by degrading collagen in the basement membrane of vessels, enabling blood vessel remodeling and growth [207]. Its role in CVDs could therefore be ambivalent and further evidence is needing in the specific setting of PAD.

Two other biomarkers to be briefly mentioned are MMP-8 and its inhibitor, namely endogenous tissue inhibitor of MMP (TIMP)-1, released from specific granules and secretory vesicles of neutrophils, respectively. Mononuclear phagocytes express MMP-8 in vitro upon stimulation with pro-inflammatory cytokines such as IL-1β, and MMP-8 mRNA and protein have been found in human atheroma [208]. Notably, MMP-8 colocalizes with cleaved but not intact type I collagen within the shoulder region of the plaque, a frequent site of rupture [208]. MMP-8 serum levels are increased in patients with CLTI undergoing lower limb revascularization compared with healthy controls, and particularly in those who experienced major amputations or death during the follow-up period [157]. In another small cohort of patients with atherosclerotic diseases including aorto-occlusive disease treated with vascular surgery, serum MMP-8, TIMP-1, and MMP-8/TIMP-1 were significantly higher in affected subjects than in healthy blood donors [158]. Similarly, others have found that circulating levels of TIMP-1 are higher in patients with PAD, and particularly in those with CLTI [156]. TIMP-1 seems to be more abundant in muscle samples of patients with ALI and CLTI than in control individuals [204]. Furthermore, calcification areas in atherosclerotic plaques are consistently associated with increased TIMP-1 expression [209], whereas mice lacking TIMP-1 demonstrate significantly increased neo-intimal formation compared with wild-type mice after femoral artery injury [210]. Of note, in Framingham Heart Study participants, total plasma TIMP-1 was correlated with major CV risk factors and indices of left ventricular hypertrophy and systolic dysfunction [211]. However, further studies are needed to better explore the role of MMPs and TIPM-1 in the natural history of PAD.

### 5.5. Neutrophil Extracellular Traps and Inflammasomes

Markers of NETs have been detected at the site of thrombus formation and plaque rupture in several CVD settings [32,33,102,212], including PAD [213].

To the best of our knowledge, the only existing study on NETs markers in patients with symptomatic, stable PAD demonstrated higher levels of human NE/α1anti-trypsin complexes but lower levels of nucleosomes in the latter than in those with non-acute deep vein thrombosis [160]. In a study on patients with PAD undergoing infra-inguinal angioplasty with stent implantation, circulating citrullinated histone H3 and cell-free DNA were measured as surrogate markers of NET formation. At 2-year follow-up, both citrullinated histone H3 and cell-free DNA were associated with the primary endpoint, defined as nonfatal myocardial infarction, stroke or transient ischemic attack, CV death, and >80% target vessel restenosis on univariate analysis [161].

Although there is no evidence on the role of NETosis in CLTI, some information has been provided in the setting of diabetic ulcers. In an elegant study by Fadini et al., expression of several NET components was significantly higher in non-healing versus rapidly healing diabetic wounds and circulating markers of NETosis were significantly higher in patients with diabetic foot syndrome compared with those without. Purified neutrophils from patients with diabetic foot syndrome were primed to undergo NETosis in vitro. Intriguingly, wound healing was delayed in mice with streptozotocin-induced diabetes compared with non-diabetic mice, whereas pretreatment with Cl-amidine, which is able to block NETosis, restored normal diabetic wound healing [214]. Further research is needed to better explore the role of NETosis in the full spectrum of PAD severity.

Likewise, exhaustive research on inflammasomes and their interplay with NETs is lacking. In humans, previous studies have shown increased expression of NLRP3 inflammasome, IL-1β, and IL-18 in patients with CAD, particularly in those with acute myocardial infarction compared with those with stable angina [90,91,92]. Serum IL-1β levels were significantly increased in patients with diabetes and PAD compared with both healthy controls and subjects with diabetes without PAD. Moreover, NLRP3 inflammasome expression was significantly increased in vascular cells of patients with diabetic foot, especially in VSMCs [162]. The platelet NLRP3 inflammasome was upregulated in mice undergoing hindlimb ischemia, which was mediated by platelet TLR4 and resulted in elevated platelet aggregation levels. Furthermore, transgenic mice with platelet-specific ablation of TLR4 and NLRP3 knockout showed improved muscle tissue perfusion after femoral artery ligation [215]. Hyperuricemia was associated with greater NLRP3 and gasdermin D expression on human atheroma plaques in patients with PAD who were candidates for amputation [216]. Lastly, human aortic ECs exposed to the plasma from patients with CLTI and treated with simvastatin responded with higher NLRP1 expression than those exposed to plasma from patients with intermittent claudication treated with simvastatin [217]. The same was described when cells were exposed to plasma from patients with PAD without previous aspirin treatment [218]. Noteworthy, both cells exposed to plasma from patients with PAD and those exposed to plasma from healthy subjects were found to undetectably express the NLRP3 gene [219]. Such findings indicate that inhibition of NLRP1 inflammasome may be a novel therapeutic approach in the context of PAD. However, the clinical relevance of these findings is uncertain. Studies providing relevant insight on inflammasome activation in neutrophils in the specific setting of PAD are awaited.

## 6. Novel Fields Worth Exploring

### 6.1. Proprotein Convertase Subtilisin/Kexin Type 9

An intriguing mediator to mention in this context is proprotein convertase subtilisin/kexin type 9 (PCSK9), a serine-protease enzyme first described in 2003 [220] and expressed by the liver, kidney and small intestine, macrophages [221], as well as by vascular cells including ECs and VSMCs [222]. While the primary effect of PCSK9 inhibitors in CV biology is mediated through the up-regulation of LDLr and the subsequent dramatic reduction in circulating LDL-C levels, mounting evidence in mice and humans suggests possible pleiotropic effects of PCSK9 [223]. In particular, PCSK9 increases LDL-C uptake by macrophages scavenger receptors, contributing to cell foam formation [224]. Furthermore, it promotes inflammation at the atherosclerotic vascular wall by inducing the expression of adhesion molecules, chemo-attractants, and inflammatory cytokines [225]. Lastly, it induces ECs apoptosis reducing vessel stability [226].

Intriguingly, PCSK9 expression levels are increased in mouse macrophages after LPS stimulation, as a result of NLRP3 inflammasome activation. In fact, NLRP3 and its downstream signals IL-1β, IL-18, and caspase-1 all participate in PCSK9 secretion, as confirmed by specific gene deletion studies [227].

Evidence on the association between PCSK9 and NETs is scarce. In a mouse model of inferior vena cava ligation, venous thrombus analysis revealed more NETs in wild type than in pcsk9-deficient mice [228]. On the other hand, the only available publication on humans is a study published in 2019 on 25 consecutive patients with acute myocardial infarction in whom neutrophils were isolated and stimulated to assess the degree of NET formation ex vivo using ionomycin and PCSK9 [229]. Surprisingly, although NETs were significantly higher in patients than in controls, the addition of PCSK9 was able to decrease ionomycin-induced NET release in a dose-dependent manner, suggesting that further studies are needed to better understand these mechanisms.

In recent times, the use of novel drugs capable of inhibiting PCSK9 in familial hypercholesterolemia and secondary CV prevention has been a source of great enthusiasm in the scientific community. Improving knowledge of the role of PCSK9 in the context of neutrophil activation and atherosclerosis would therefore have crucial therapeutic repercussions.

### 6.2. Exercise Programs and Neutrophil Biology

The main treatments for PAD include exercise programs, pharmacological strategies, and arterial revascularization. The primary objective of conservative therapy is to reduce the risk of MACE and MALE. Therapeutic strategies for PAD also seek to improve symptoms, functional capacity, and overall quality of life in affected patients. Exercise interventions are highly recommended in this setting as first-line treatment (class of recommendation: I, level of evidence: (A) [230]. In particular, supervised exercise programs (SEPs) have been shown to be superior to unsupervised exercise (“home-exercise” programs, HEPs) with in improving treadmill performance [231] and promoting physiological changes in the lower limbs [232]. However, a systematic review [233] provided some evidence that HEPs may also improve walking distances and quality of life in patients with intermittent claudication.

Despite these beneficial effects, relatively few studies to date have evaluated the biological mechanisms responsible for these improvements. Several hypotheses have been raised in this regard, such as an improved muscle mitochondrial energy metabolism and endothelial function through stimulation of nitric oxide-dependent vasodilation, increased perfusion and oxygenation of the lower limbs, reduced systemic inflammation, as well as change in gait pattern [234]. Among all these mechanisms, modulation of inflammatory responses appears to be a promising element to explore. In fact, previous research by our group using a mouse model of PAD showed that running wheel training modulates the phenotype of circulating monocytes towards an anti-inflammatory profile. Moreover, training inhibits inflammation and pro-inflammatory macrophage polarization in the lower limb ischemic muscle [235]. The positive effects of physical activity on inflammation have also been widely demonstrated in various human populations [236,237,238]. In patients with PAD, higher levels of physical activity were independently associated with lower levels of inflammatory markers [239,240], whereas several trials have shown beneficial effects of exercise programs in terms of systemic inflammation [241,242]. Previous research has found that both SEPs and HEPs are effective in reducing cultured EC apoptosis in patients with symptomatic PAD, while SEPs elicits additional vascular benefits by improving circulating markers of endogenous antioxidant capacity, angiogenesis, endothelium-derived inflammation, and blood glucose concentration in patients with symptomatic PAD [241,243,244,245]. However, to the best of our knowledge, no study has so far evaluated the effects of SEPs and HEPs on neutrophil biology including degranulation, NETs generation, and inflammasome activation in patients with symptomatic PAD, nor has it assessed whether there is a correlation between the change in the levels of the aforementioned markers and improvement of parameters of walking functional capacity in these patients. Clarifying such aspects could shed light on the biology underlying the effects of physical activity in PAD. Furthermore, identifying predictors of successful treatment response among markers of neutrophil activation could help to tailor exercise training in patients with PAD. Further research should clarify these intriguing aspects.

## 7. Conclusions

As accumulating evidence supports a central role of neutrophils and their mediators in CVD, further research is needed to elucidate their etiopathogenic role and their potential value as diagnostic and prognostic markers across the full spectrum of PAD severity. In the future, complementary assessment of neutrophil activity including novel mechanisms, such as NETs generation and inflammasome pathways, should improve risk stratification and provide personalized treatments to patients with PAD.

## Figures and Tables

**Figure 1 ijms-24-01169-f001:**
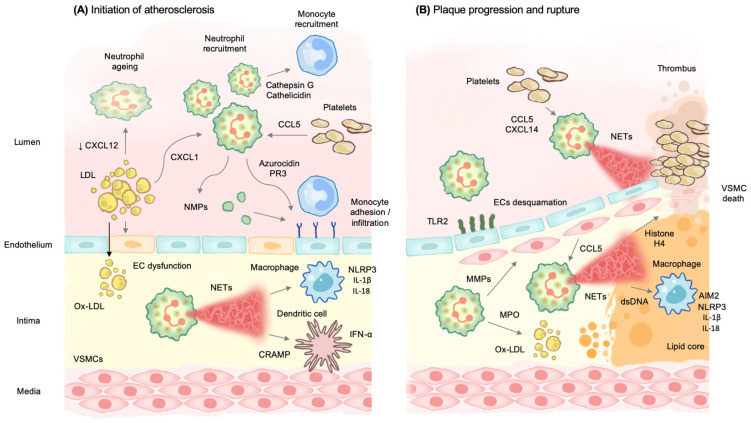
The role of neutrophils in atherosclerosis. (**A**) During the first stages of atherosclerosis, unbalanced blood lipid profiles, abnormal shear forces, and local release of pro-inflammatory cytokines cause EC dysfunction with subsequent accumulation of ox-LDL in the intima. Moreover, hypercholesterolemia increases neutrophil mobilization through CXCL1 while decreasing neutrophils clearance in the bone marrow and enhancing their aging through reduced CXCL12 levels. Platelet-derived CCL5 also promotes neutrophil activation and recruitment. Activated neutrophils secrete cathepsin G and cathelicidin that enhance monocyte recruitment. Other granule proteins, such as azurocidin and PR3, increase endothelial expression of adhesion molecules and regulate EC permeability. NMPs further activate immune cells and promote cytokine release and adhesion protein expression in ECs. Once activated, neutrophil undergo NETosis. NETs containing CRAMP complexes stimulate plasmacytoid dendritic cells to produce IFN-α and induce macrophages to produce IL-1β and IL-18 via the NLRP3 inflammasome. These mechanisms lead to chronic inflammation and initiate atherosclerosis. (**B**) VSMCs begin to proliferate and migrate from the media to the intima, where they participate in the formation of the fibrous cap that covers the inflammatory core of the plaque. In more advanced stages of disease, neutrophils induce VSMC death via MMPs release. Moreover, activated VSMCs stimulate NETs generation through the secretion of CCL7, and histone H4 has cytotoxic effects in VSMCs. Platelets also promote NETs generation through CCL5 and CXCL4. Extracellular dsDNA colocalizes in mature atherosclerotic lesions with the noncanonical inflammasome AIM2 and stimulate macrophages to produce IL-1β and IL-18 via the NLRP3 inflammasome. Neutrophils trigger luminal ECs desquamation, a process known as plaque erosion, by stimulating TLR2 on ECs.

**Table 1 ijms-24-01169-t001:** Clinical studies on the role of biomarkers of neutrophil activation in the setting of lower limb peripheral artery disease.

Assessed Biomarker	Role of the Biomarker in PAD	Summary of the Studies	Ref.
**NLR**	Diagnosis	NLR levels are significantly higher in patients with PAD than in healthy subjects	[116,117]
	Severity stratification	NLR levels are significantly associated with severe multi-level PAD and CLTI presence	[116,118,119,120,121,122,123]
	Short-term prognosis after lower limb revascularization	NLR levels are independently associated with in-hospital death, MACE, MALE, renal failure, and length of hospitalization	[124,125]
	Long-term prognosis after lower limb revascularization	NLR levels are independently associated with all-cause death, MACE, MALE, patency loss/graft failure up to 10 years after the procedure	[123,125,126,127,128,129,130,131,132,133,134,135]
	Prognosis in patients with CLTI unsuitable for revascularization	NLR levels are independently associated with poor response to conservative treatment and MALE, as well as death at 3-year follow-up	[136,137,138]
	Prognosis in patients with ALI after lower limb revascularization	NLR levels are independently associated with both death and amputation at 30 days and up to 8 years after the procedure	[139,140,141,142,143]
**MPO**	Diagnosis	MPO levels are significantly higher in patients with PAD than in healthy subjects	[144,145,146]
	Severity stratification	MPO levels are independently and inversely associated with ABI levels. Furthermore, MPO levels are significantly higher in patients with two-bed vascular disease compared with PAD alone	[145,147,148]
	Prognosis	MPO levels are independently associated with MACE and MALE occurrence at 3.3-year follow-up	[148,149,150]
**NGAL**	Diagnosis	NGAL and uNGAL/uCr levels are significantly higher in patients with PAD than in healthy subjects. NGAL levels are significantly higher in patients with ALI than in those without it	[151,152,153]
	Prognosis	MPO levels are independently associated with MACE and MALE occurrence	[151,154]
**MMP-8** **MMP-9** **TIMP-1**	Diagnosis	MMP-9, MMP-8, TIMP-1, and MMP-8/TIMP-1 levels are significantly higher in patients with PAD than in healthy subjects. MMP-9 levels are significantly higher in patients with ALI than in those without it	[155,156,157,158]
	Severity stratification	MMP-9 and TIMP-1 are significantly higher in patients with CLTI than in those with intermittent claudication	[156]
	Long-term prognosis after lower limb revascularization	Post-operative increase in MMP-9 levels is significantly associated with graft occlusion at 9-month follow-up	[159]
**NETs**	Diagnosis	Nucleosome levels are significantly lower and human NE/α1anti-trypsin levels are significantly higher in patients with PAD compared with patients with deep vein thrombosis	[160]
	Long-term prognosis after lower limb revascularization	Citrullinated histone H3 and cell-free DNA levels are independently associated with MACE and > 80% target vessel restenosis at 2-year follow-up	[161]
**Inflammasome**	Diagnosis	IL-1β levels are significantly higher in patients with PAD and diabetes than in healthy subjects and patients with diabetes without PAD	[162]

## Data Availability

Not applicable.

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
