# Peer review of "The Role of Neutrophils in Lower Limb Peripheral Artery Disease: State of the Art and Future Perspectives"

_ijms, 2023, doi:10.3390/ijms24021169_

Round 1

Reviewer 1 Report

Buso et al presents a narrative review paper aiming to discuss the interplay between neutrophils biology and lower limb peripheral artery disease.

I found the paper well-written, nicely organized as well as discusses an important area in the medical field. Moreover, the review of the literature is truly comprehensive, thus the paper is more valuable.

I have some majors that should be fixed during the review process:

I would like the introduction of main figure that presents the role of neutrophils into PAD pathogenesis at glance - it will bring more attractiveness to the readers of the paper. Right now, one image might show more than 1000 words.

Moving further, there is a very limited amount of information given regarding the following PRISMA guidelines and description of the searching strategy, which is so crucial for review papers, even narrative reviews. The Author should include at least: Data sources and searches, Study eligibility criteria, Study selection process, Data extraction, and study quality assessment (assessing the risk of bias (ROB) for each included study), Data synthesis. MeSH terms (in addition/replacement of keywords) are necessary to be included. For each step, it is necessary to explain to the reader with pictures or tables. It is necessary to explain what was drawn at each step to lead to the result. Moreover, a figure showing the PRISMA-based workflow must be drawn accordingly to the Prisma schema. After that, a discussion is valuable even for narrative papers. Description of Data Mining strategy should also be included.

Moreover, since the year of 2022 brought some new very significant discoveries in neutrophils biology and NETs formation, I would like to see these papers discussed (all of them come from top-rated journals and are considered as dogma-shift discoveries):

Montaldo, E., Lusito, E., Bianchessi, V. et al. Cellular and transcriptional dynamics of human neutrophils at steady state and upon stress. Nat Immunol 23, 1470–1483 (2022). https://doi.org/10.1038/s41590-022-01311-1

Ravi Vats, Tomasz W. Kaminski, Tomasz Brzoska, John A. Leech, Egemen Tutuncuoglu, Omika Katoch, Jude Jonassaint, Jesus Tejero, Enrico M. Novelli, Tirthadipa Pradhan-Sundd, Mark T. Gladwin, Prithu Sundd; Liver-to-lung microembolic NETs promote gasdermin D–dependent inflammatory lung injury in sickle cell disease. Blood 2022; 140 (9): 1020–1037. doi: https://doi.org/10.1182/blood.2021014552

Russo, A.J., Vasudevan, S.O., Méndez-Huergo, S.P. et al. Intracellular immune sensing promotes inflammation via gasdermin D–driven release of a lectin alarmin. Nat Immunol 22, 154–165 (2021). https://doi.org/10.1038/s41590-020-00844-7

Please also somehow format the table - it is very painful to scrool right and left the whole table.

There is some minor grammatical error / minor typos - please go over it carefully.

Author Response

We are grateful to the Reviewer for the comments, whose relevance helped us to further improve the quality of the manuscript.

Attached please find the marked and clean versions of the revised manuscript “R1”. In addition, below please find are our response to the Reviewer’s comments.

Buso et al presents a narrative review paper aiming to discuss the interplay between neutrophils biology and lower limb peripheral artery disease.

I found the paper well-written, nicely organized as well as discusses an important area in the medical field. Moreover, the review of the literature is truly comprehensive, thus the paper is more valuable.

We are very thankful to the Reviewer for this kind remark.

 I have some majors that should be fixed during the review process:

  • I would like the introduction of main figure that presents the role of neutrophils into PAD pathogenesis at glance - it will bring more attractiveness to the readers of the paper. Right now, one image might show more than 1000 words.

We fully agree with Reviewer's proposal. Accordingly, we have added a figure summering the role of neutrophil activation in the setting of atherogenesis and plaque progression and rupture (Figure 1). 

  • Moving further, there is a very limited amount of information given regarding the following PRISMA guidelines and description of the searching strategy, which is so crucial for review papers, even narrative reviews. The Author should include at least: Data sources and searches, Study eligibility criteria, Study selection process, Data extraction, and study quality assessment (assessing the risk of bias (ROB) for each included study), Data synthesis. MeSH terms (in addition/replacement of keywords) are necessary to be included. For each step, it is necessary to explain to the reader with pictures or tables. It is necessary to explain what was drawn at each step to lead to the result. Moreover, a figure showing the PRISMA-based workflow must be drawn accordingly to the Prisma schema. After that, a discussion is valuable even for narrative papers. Description of Data Mining strategy should also be included.

We thank the Reviewer for the suggestion. However, although we agree to introduce some necessary information on search methodology, we respectfully disagree with part of the request. We propose to add a paragraph regarding Data sources and search strategy with MeSH terms (page 4, lines 14-25, and page 5, lines 1-2). The PRISMA (Preferred Reporting Items for Systematic Reviews and Meta-Analyses) recommendations, on the other hand, refer precisely to Systematic Reviews and Meta-Analyses only. Accordingly, introducing aspects such as Study eligibility criteria, Study selection process, Data extraction, Study quality assessment (assessing the risk of bias for each included study), Data synthesis, PRISMA-based workflow, and Data Mining strategy into the manuscript might be methodologically inappropriate, as our is a narrative review. Moreover, our review covers not only studies conducted on neutrophils in PAD, but also classical and innovative aspects of neutrophil pathophysiology and atherosclerosis, which would further make it difficult to apply PRISMA recommendations to our work. We hope the Reviewer will understand our objection.

  • Moreover, since the year of 2022 brought some new very significant discoveries in neutrophils biology and NETs formation, I would like to see these papers discussed (all of them come from top-rated journals and are considered as dogma-shift discoveries):
    • Montaldo, E., Lusito, E., Bianchessi, V. et al. Cellular and transcriptional dynamics of human neutrophils at steady state and upon stress. Nat Immunol 23, 1470–1483 (2022). https://doi.org/10.1038/s41590-022-01311-1
    • Ravi Vats, Tomasz W. Kaminski, Tomasz Brzoska, John A. Leech, Egemen Tutuncuoglu, Omika Katoch, Jude Jonassaint, Jesus Tejero, Enrico M. Novelli, Tirthadipa Pradhan-Sundd, Mark T. Gladwin, Prithu Sundd; Liver-to-lung microembolic NETs promote gasdermin D–dependent inflammatory lung injury in sickle cell disease. Blood 2022; 140 (9): 1020–1037. doi: https://doi.org/10.1182/blood.202101455.
    • Russo, A.J., Vasudevan, S.O., Méndez-Huergo, S.P. et al. Intracellular immune sensing promotes inflammation via gasdermin D–driven release of a lectin alarmin. Nat Immunol 22, 154–165 (2021). https://doi.org/10.1038/s41590-020-00844-7

We appreciate the Reviewer’s suggestion. We have added the above articles to the relevant parts of the manuscript (page 4, line 6; page 13, line 21; and page 7, line 7, respectively).

  • Please also somehow format the table - it is very painful to scrool right and left the whole table.

We have tried formatting the tables and summarize them into one to make data easier to read, as suggested.

  • There is some minor grammatical error / minor typos - please go over it carefully.

We thank the Reviewer for the comment. We have reviewed the text extensively and corrected any grammatical errors and typos throughout the manuscript.

Reviewer 2 Report

In the submitted review article titled, “What role for neutrophils in lower limb peripheral artery disease? State of the art and future perspectives” authors Buso et. al., reviewed literature on the role of neutrophil in cardiovascular disease, discussed the implications of neutrophil activity in lower limb peripheral artery disease, presented recent advances and future perspectives hence suffice the knowledge gap. My comments are as follows;

1.       “Recent advances in understanding neutrophil biology” should come before “Novel fields worth exploring”- This subtitle will get benefited by including topic like “Neutrophil plasticity” or “Regulation of neutrophil metabolism” in CVD/PAD. Also, author should consider discussing articles published in last 5 years to present “Recent advances”.

2.       “Role of neutrophil biomarkers in peripheral artery disease”. In this column the biomarkers discussed are common in cardiovascular disease. It will be important to highlight biomarkers/ functions/signaling or neutrophils that are unique in PAD in comparison to other cerebrovascular disease.

3.       Abbreviations should appear along with the text. For example, MALE first appears on page 7, line 310, but it is mentioned on page 21 under the table.

4.       Table 1 and 2 – Author should concise the information and make it easier to read and compare.   

5.       Summary Diagram showing the role of neutrophil biomarkers in peripheral artery disease is highly recommended.

6.       “What role for neutrophils” in the title, is grammatically incorrect.

Author Response

We are grateful to the Reviewer for the comments, whose relevance helped us to further improve the quality of the manuscript.

Attached please find the marked and clean versions of the revised manuscript “R1”. In addition, below please find are our response to the Reviewer’s comments.

In the submitted review article titled, “What role for neutrophils in lower limb peripheral artery disease? State of the art and future perspectives” authors Buso et al. reviewed literature on the role of neutrophil in cardiovascular disease, discussed the implications of neutrophil activity in lower limb peripheral artery disease, presented recent advances and future perspectives hence suffice the knowledge gap. My comments are as follows;

  • “Recent advances in understanding neutrophil biology” should come before “Novel fields worth exploring”- This subtitle will get benefited by including topic like “Neutrophil plasticity” or “Regulation of neutrophil metabolism” in CVD/PAD. Also, author should consider discussing articles published in last 5 years to present “Recent advances”.

We thank the Reviewer for these helpful suggestions. We have added references of articles that have been published in recent years and introduced two additional paragraphs ("Neutrophil heterogeneity and plasticity" and "Regulation of neutrophil metabolism") at the end of the section "Recent advances in understanding neutrophil biology" (page 8, lines 18-24, and page 9, lines 1-17).

However, as this section includes evidence that has emerged not only recently, but in the last decade, we have replaced the section title with "Issues challenging the old conception of neutrophil biology."

Lastly, although we agree in principle with the Reviewer with the fact that this section should be moved at the end, before "Novel fields worth exploring", it actually introduces the following sections, which would be therefore difficult to understand without some basics about NETosis and inflammasomes given here. Therefore, we kindly ask that this section be left in its current position, hoping that the Reviewer will agree to our request.

  • “Role of neutrophil biomarkers in peripheral artery disease”. In this column the biomarkers discussed are common in cardiovascular disease. It will be important to highlight biomarkers/ functions/signaling or neutrophils that are unique in PAD in comparison to other cerebrovascular disease.

We thank the Reviewer for highlighting this important aspect. Unfortunately, the studies conducted in the context of PAD involve biomarkers that have already been extensively studied in the context of other common CVDs, whereas no biomarkers specific to PAD have emerged with respect to the latter. This lack of knowledge is a major limitation in the understanding of the natural history of PAD and will certainly need to be addressed in the future. We have added a sentence to clarify these issues at the beginning of the " The established role of neutrophils in peripheral artery disease" section (page 14, lines 9-16).

  • Abbreviations should appear along with the text. For example, MALE first appears on page 7, line 310, but it is mentioned on page 21 under the table.

We thank the Reviewer for the comment. We have reviewed the text extensively and corrected any abbreviations throughout the manuscript.

  • Table 1 and 2 – Author should concise the information and make it easier to read and compare.

We have tried formatting the tables and summarize them into one to make data easier to read, as suggested.

  • Summary Diagram showing the role of neutrophil biomarkers in peripheral artery disease is highly recommended.

We fully agree with Reviewer's proposal. Accordingly, we have added a figure summering the role of neutrophil activation in the setting of atherogenesis and plaque progression and rupture (Figure 1).

  • “What role for neutrophils” in the title, is grammatically incorrect

We thank the Reviewer for the comment. We have modified the title accordingly.

Round 2

Reviewer 1 Report

The Authors made a review process in a satisfactory way and right now, I feel that the paper fits well the journal. The Authors correctly addressed all my majors and minors. I am truly happy with this review process.